# Dental Service Provision and Oral Health Conditions of Children Aged 0–12 Years, Northern Thailand: Transferring of Sub-District Health Promotion Hospitals Policy Era

**DOI:** 10.3390/healthcare13080874

**Published:** 2025-04-11

**Authors:** Noppcha Singweratham, Anon Fuengkhajorn, Jukkrit Wungrath, Pallop Siewchaisakul

**Affiliations:** 1Faculty of Public Health, Chiang Mai University, Chiang Mai 50200, Thailand; noppcha.s@cmu.ac.th (N.S.); jukkrit.w@cmu.ac.th (J.W.); 2Dental Department, Bo Kluea Hospital, Nan 55220, Thailand; anon.anonymous57190271@gmail.com

**Keywords:** dental health services, oral health, child, local government, Thailand

## Abstract

**Background/Objectives:** Some of the sub-district health promotion hospitals (SHPHs) have transferred to local administrative organizations (LAOs). This may impact dental services and oral health of children aged 0–12 years. This study aimed to investigate and compare the dental services and oral health conditions of children aged 0–12 years between transferred and non-transferred SHPHs in northern Thailand. **Methods:** The study was a retrospective cohort study. Data on dental service provision and oral health conditions were retrieved using secondary data from the national health security office between 2017 and 2021. The data were analyzed and presented using descriptive statistics, Chi-square test, and unpaired *t*-test. **Results:** Overall, the findings revealed that transferred SHPHs generally provided fewer dental services than non-transferred SHPHs. Regarding oral health conditions, transferred SHPHs reported a statistically significantly higher prevalence of dental caries compared to non-transferred SHPHs at the age of 3 years (31.3% vs. 22.2%), the age of 5 years (51.2% vs. 30.7%), and the age of 12 years (37.1% vs. 30%), respectively. Transferred SHPHs have higher decayed, missing, and filled teeth mean scores compared to non-transferred SHPHs in children aged 3 years (mean diff: 0.5; 95% CI: 0.3 to 0.7) and 12 years (mean diff: 0.3; 95% CI: 0.1 to 0.5). **Conclusions:** Transferred SHPHs have lower dental service provision but higher oral health problems than non-transferred SHPHs. Close monitoring for the dental service provision and oral health among children aged 0–12 years is needed, especially in the area of responsibility of the transferred SHPHs.

## 1. Introduction

The Universal Health Coverage system (UHC) in Thailand is one of the main privileges for Thais. Sub-district health promotion hospitals (SHPHs) are primary healthcare units in Thailand that provide essential medical services, health promotion, disease prevention, and community health programs, mainly in rural areas. They play a crucial role in the country’s UHC. In 2008, the Decentralization Action Plan (2nd Edition) was introduced, which mandates the transfer of primary healthcare services from SHPHs to local administrative organizations (LAOs), in which LAOs are decentralized government bodies responsible for local governance, including public health. They often support SHPHs by funding healthcare initiatives and managing community health programs to improve local well being. However, as of 2020, only 65 out of 9787 SHPHs nationwide have been transferred, with most transfers occurring in northern Thailand. These transfers involve sub-district administrative organizations (SAOs) and municipalities [1].

SHPHs are primary care units providing health services under the guidelines established by the Primary Health Care Act, B.E. 2562 (2019). They offer health promotion, medical treatment, disease prevention, and health rehabilitation to ensure equitable and accessible healthcare services for all individuals at any age, from birth to death, without financial barriers [2,3,4,5]. The transfer of SHPHs to LAOs has resulted in significant changes to the management and operation of these facilities. These changes include adjustments in operational processes, equipment, drug and non-drug supplies, staffing levels, and health services. While some transfers have improved local healthcare management and aligned with community needs, others have faced challenges, such as budget mismanagement and insufficient understanding of regulations by primary care units (CUPs), leading to disparities in healthcare service allocation between transferred and non-transferred SHPHs [6,7,8,9]. It can be seen that the mechanism of management and resource allocation plays a crucial role in transforming the healthcare system to promote fairness and equity. This is essential for reducing disparities in access to healthcare services [10].

Dental public health services have also been impacted by these transfers. Dental services in SHPHs, primarily provided by dental health officers/dental hygienists, are governed by the Ministry of Public Health’s regulations B.E. 2539 (1996) and supervised by dentists under the Dental Profession Act, B.E. 2537 (1994). Before the transfer, these services were supported by supervisory dentists from affiliated hospitals. However, post-transfer, the lack of oversight and support from the Ministry of Public Health has limited the ability of dental hygienists to perform their duties effectively, potentially affecting dental service provision and oral health outcomes, particularly for children aged 0–12 years [11,12]. Children aged 0–12 years represent a critical age group for oral health promotion and caries prevention [13]. Recognizing the importance of this issue, the Ministry of Public Health has established key performance indicators for this group, including oral health examinations, topical fluoride applications, dental sealants, dental caries prevalence, as well as decayed, missing, and filled teeth indices [14].

Previous studies have highlighted challenges in achieving health service targets and indicators in transferred SHPHs and in the allocation of budgets and resources between transferred and non-transferred SHPHs [15,16]. However, most research has focused on overall health service systems, access to healthcare, public satisfaction, and community participation, with limited attention to dental public health services. This study aims to investigate and compare the dental services and oral health conditions of children aged 0–12 years between transferred and non-transferred SHPHs in northern Thailand

## 2. Materials and Methods

### 2.1. Research Methodology

This study employed a quantitative research method using a retrospective cohort study design.

### 2.2. Population and Sample

This study determined population and sample based on the previous published article (Thai language) by Fuengkhajorn, A., Singweratham, N., Siewchaisakul, P., and Wungrath, J. (2024), which adopted a mixed-methods study design. The details of the population and sample were published elsewhere [12].

In brief, the population was the sub-district health promotion hospitals (SHPHs) in northern Thailand (Health Zones 1, 2, and 3) providing joint healthcare services with primary care units (PCUs), totaling 2098 facilities.

Sample: SHPHs providing dental services in collaboration with CPUs in the provinces of Kamphaeng Phet, Chiang Rai, Chiang Mai, Tak, Nan, and Phitsanulok. These SHPHs include both transferred and non-transferred groups to local administrative organizations (LAOs) before 2017. Sampling was conducted using purposive sampling based on the following criteria: SHPHs were selected from the same province to ensure comparability. The selected SHPHs were of the same size and served populations of similar demographics within their responsible areas, as shown in Table 1.

Secondary data on dental services and oral health conditions between 2017 and 2021 were recorded in the data system of sub-district health-promoting hospitals (SHPHs).

The sample was purposively selected solely from secondary data, focusing on dental services and oral health conditions in children aged 0–12 years between 2017 and 2021.

### 2.3. Research Instruments

Data Recording Forms: Derived from the SHPH data system, they are used to extract and document secondary data on dental services and oral health conditions of children aged 0–12 years.

### 2.4. Data Source and Data Collection Process

This study retrieved data on dental services and oral health conditions in children aged 0–12 years between 2017 and 2021 from the SHPH database. The data were directly recorded and documented by dental personnel (dentists or dental nurses) from SHPHs. Only registered dental personnel who log into the recording system are authorized to enter data. This ensures that the recorded data accurately reflect the actual number of dental services provided, distinguishing it from the Ministry of Public Health’s indicator data (Health data center). All data were electrically recorded. We recruit only records that are complete and follow procedure codes/diagnosis codes of the following:-Oral health examination (Procedure code: 2330011)/Diagnosis code: Z01.2-Topical fluoride application (procedure codes: 2377020, 2377021)/diagnosis codes: K02.0 (initial caries) and K03.8 (other specified diseases of the hard tissues of teeth)-Pit and fissure sealant application on permanent teeth (procedure codes: 2387030, 238703A, 238703B, 238703C, 238703D, 238703E, 238703F, 238703G, 238703H)/diagnosis codes: K02.0 (initial caries) and K02.3 (arrested dental caries)

We submitted official requests, sent formal letters to the Provincial Public Health Office and SHPH directors, and coordinated with relevant staff to access the SHPH database. All data were retrieved using record forms. We validated and ensured data completeness after extracting the data from databases.

### 2.5. Data Analysis

After collecting and verifying the accuracy and completeness of the data, the information was coded and analyzed using computer software. The analysis included the following steps:1.Analysis of dental services and oral health conditions

Dental services and oral conditions were presented as frequency and percentage. Graphs were used to observe data trends.

2.Comparison of dental services and oral health conditions between transferred and non-transferred SHPHs. Trends were visualized using graphs. Statistical analysis included the Chi-square test and unpaired *t*-test to identify differences between the groups.

### 2.6. Research Ethics

This study was approved by the Human Research Ethics Committee of the Faculty of Public Health, Chiang Mai University, under approval number ET039/2023, dated 15 September 2023.

## 3. Results

The general context of transferred and non-transferred sub-district health-promoting hospitals (SHPHs) was found to be similar. Key factors included distance from the parent hospital, population within the service area, number of schools in the service area, number of child development centers in the service area, and position of the dental public health officer responsible for providing dental services at the SHPH. These findings are summarized in Table 2.

### 3.1. Comparison of Dental Services for Children Aged 0–12 Years

The overall provision of dental services for children aged 0–12 years, averaged over the period from 2017 to 2021, revealed that transferred sub-district health-promoting hospitals (SHPHs) provided fewer dental services compared to non-transferred SHPHs. An exception was noted in 2021, where oral health examinations for children aged 0–12 years were more frequently conducted by the transferred SHPHs. These findings are presented in Table 3.

The trend in dental services for children aged 0–12 years showed a decline for both transferred and non-transferred sub-district health promoting hospitals (SHPHs), reaching the lowest point in 2021. However, an exception was observed in oral health examinations for children aged 0–12 years in transferred SHPHs, where the trend showed an increase. These trends are illustrated in Figure 1.

### 3.2. Comparison of Oral Health Conditions for Children Aged 0–12 Years

The overall oral health conditions of children aged 0–12 years, averaged over the period from 2017 to 2021, showed that non-transferred sub-district health-promoting hospitals (SHPHs) had lower prevalence rates of dental caries and lower averages of decayed, filled, and extracted teeth compared to transferred SHPHs. These findings are detailed in Table 4.

Additionally, it was found that the overall oral health conditions between the groups of SHPHs that were transferred and those that were not transferred, including the prevalence of dental caries in children aged 3, 5, and 12 years, as well as the average number of decayed, filled, or extracted teeth in children aged 3 and 12 years, showed statistically significant differences, except for the average number of decayed, filled, or extracted teeth in the 5-year-old group, which were not different, as shown in Table 5.

From Figure 2, the overall prevalence of dental caries in the non-transferred SHPHs tends to decrease, while in the transferred SHPHs, the majority tends to increase.

## 4. Discussion

Dental services have been listed in the benefit package that provides free of charge dental care for Thai children aged 0–12 years old. Recently, a policy of transferring SHPHs from the Ministry of Public Health to LAOs has been introduced in Thailand, which changes resource allocation and may impact dental services and oral health, especially among children aged 0–12 years old. This study investigated and compared the dental services and oral health conditions of children aged 0–12 years between transferred and non-transferred SHPHs in northern Thailand. The study’s main findings were consistent with the alternative hypothesis that transferred SHPHs provided fewer dental services than non-transferred SHPHs. The prevalence of dental caries and mean scores for decayed, missing, and filled teeth were found to be higher in transferred SHPHs than in the non-transferred SHPHs.

The previous evidence revealed that there are various problems after SHPHs transferred to LAOs, including health care service providing, disease prevention, and health care information [7]. Based on the study’s findings, it revealed that during 2017–2021, it was found that SHPHs that had been transferred provided fewer dental services than those that had not been transferred across all types of dental services. These services included oral health examinations for children aged 0–12 years, fluoride varnish application for children aged 0–5 years, and dental sealants for permanent teeth in 6-year-old children. This aligns with evaluations of the transfer of primary healthcare services, which showed a similar trend in other health services, such as screening for diabetes, hypertension, and cervical cancer [9]. The transfer of responsibilities may lead to confusion about work scope and fear among some dental public health officers in providing dental services. Additionally, there is a lack of systematic support from parent hospitals regarding dentists, equipment, tools, medications, and non-medical supplies. This has resulted in disparities between transferred and non-transferred SHPHs [16].

Furthermore, the focus of SHPHs is on providing general, non-complex basic dental services. Additionally, proactive services are offered in schools and early childhood development centers within their jurisdiction. Under the Dental Profession Act B.E. 2537 [17], the certification of dental public health officers’ work differs. Transferred HPHs often lack clarity regarding certification details and the list of supervising dentists, leading to confusion about the scope of work and fear among dental public health officers. This situation aligns with policy recommendations regarding the transfer of HPHs to provincial administrative organizations (PAOs) [15] and budget allocation disparities in the National Health Security Fund for primary healthcare providers [18]. Studies have highlighted differences in health service delivery and resource allocation [7]. Therefore, it is essential to establish equitable support and resource allocation models for both transferred and non-transferred SHPHs. This approach would ensure fairness and reduce disparities in public access to dental healthcare services [3,4].

In addition, based on our published article, however, in Thai language [12], transferred SHPHs face challenges such as the lack of rotating dentists to provide support and insufficient provision of equipment, tools, medications, and non-medical supplies from parent hospitals. These limitations may reduce the capacity of transferred HPHs to provide dental services [12]. Moreover, the access to dental services in oral examination in children aged 0–12 years, fluoride application in children aged 0–5 years, and dental sealant on permanent teeth in children aged 6 years were statistically different between those two types of the SHPHs. This is because dental public health officers are required to provide services under the supervision of a dentist, as stated in the act, which specifies that dental public health officers are not licensed as dental practitioners.

In the overview of oral health status in children aged 0–12 years: on average, during 2017–2021, it was found that sub-district health promotion hospitals (SHPHs) that had been transferred reported higher prevalence rates of dental caries and higher mean scores for decayed, missing, and filled teeth (DMFT) across all studied age groups compared to SHPHs that had not been transferred. Statistical analysis showed significant differences, except for the mean DMFT scores among 5-year-old children, where no statistically significant differences were observed between transferred and non-transferred SHPHs. This is consistent with certain performance indicators set by the Ministry of Public Health for transferred HPHs managed by local administrative organizations [15]. The prevalence of dental caries in children may be influenced by various factors, such as the consumption of starchy and sugary foods, improper oral hygiene practices, lack of knowledge about oral health, irregular visits to the dentist, and other factors related to the children’s parents [13]. The transfer of responsibilities might, therefore, have an unclear impact on the oral health status of children across all age groups. Moreover, when analyzing the trends in dental caries prevalence, it was found that the overall prevalence in non-transferred SHPHs tended to decrease, whereas transferred SHPHs showed an increasing trend. As a result, differences in oral health status between transferred and non-transferred SHPHs may become more pronounced over time. Notably, the dental caries prevalence in both transferred and non-transferred SHPHs was lower than the average of the world, which reported the prevalence of dental caries in primary teeth in children in the world was 46.2%, and the prevalence of dental caries in permanent teeth in children in the world was 53.8% [19]. However, the dental caries and DMFT for age 5- year-old in this present study, especially in transferred SHPHs was higher than a report and projection of a study in southern Thailand [20]. Therefore, SHPHs in northern Thailand should focus to monitor the oral health problem focus on children age of 5 years.

The present study did have some limitations, this study uses secondary data from SHPHs; hence, other characteristics or factors that are related to health conditions are limited. The data used were solely from records in the SHPHs database. Therefore, to obtain more comprehensive data on dental service provision for children aged 0–12 years, further studies should incorporate additional databases, such as those from schools, early childhood development centers, and claim data from the national health security office. Further research should also examine the costs of dental services and explore dental care for other population groups within transferred SHPHs. In addition to the limitations, the present study has its strengths of demonstrating the dental service and oral health using the most updated data from SHPHs, which represent the real situation of both transferred and non-transferred SPHPs.

## 5. Conclusions

Transferred SHPHs have lower dental service provision but higher oral health problems than non-transferred SHPHs. Close monitoring for the dental service provision and oral health among children aged 0–12 years is needed, especially in the area of responsibility of the transferred SHPHs. The Ministry of Public Health and Ministry of Interior should revise resource allocation, including personnel, equipment, and supplies, to ensure fairness and consistency in service delivery for transferred SHPHs.

## Figures and Tables

**Figure 1 healthcare-13-00874-f001:**
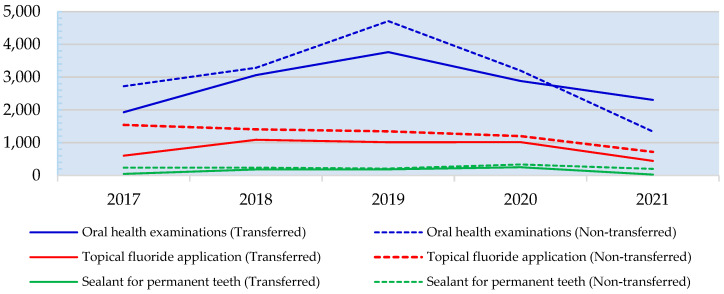
Trends in dental services for children aged 0–12 years during 2017–2021.

**Figure 2 healthcare-13-00874-f002:**
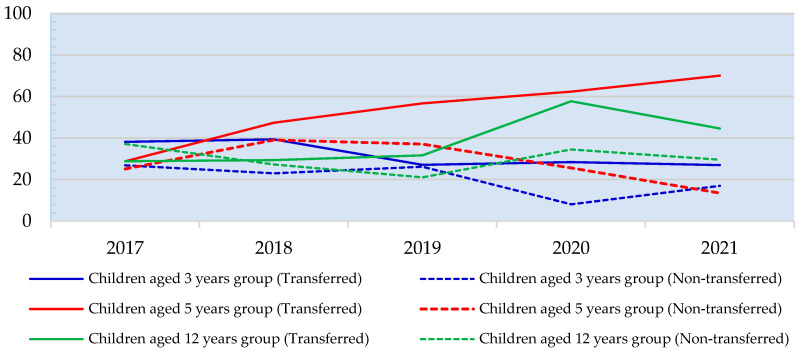
Trends in the prevalence of dental caries among children aged 0–12 years during 2017–2021.

**Table 1 healthcare-13-00874-t001:** Summarizes the selection criteria and distribution of SHPHs across the study provinces.

Province	Transfer status	Setting	Size	Population in the Area of Responsibility (People)
Kamphaeng Phet	Transferred	SHPH A	S	2632
	Non-transferred	SHPH B	S	2482
Chiang Rai	Transferred	SHPH C	M	3816
	Non-transferred	SHPH D	M	5410
Chiang Mai	Transferred	SHPH E	M	6753
	Non-transferred	SHPH F	M	7518
Tak	Transferred	SHPH G	M	4545
	Non-transferred	SHPH H	M	4781
Nan	Transferred	SHPH I	M	3861
	Non-transferred	SHPH J	M	3842
Phitsanulok	Transferred	SHPH K	L	10,146
	Non-transferred	SHPH L	L	13,802

Abbreviations: SHPH, sub-district health promotion hospital.

**Table 2 healthcare-13-00874-t002:** The general context of transferred and non-transferred sub-district health-promoting hospitals (SHPHs).

General Context	Status		*p*-Value
	Transferred SHPH Group (Mean, SD)	Non-Transferred SHPH Group (Mean, SD)	
**Distance from network hospital (kilometers)**	10.6 ± 4.7	10.1 ± 4.7	0.871 ^#^
**Population in the responsible area (people)**	5320.0 ± 2604.5	5320.0 ± 2604.5	0.632 ^#^
**Schools in the responsible area (institutions)**	3.5 ± 1.0	3.0 ± 1.1	0.438 ^#^
**Child Development Centers in the responsible area (centers)**	1.3 ± 1.2	1.8 ± 1.8	0.484 ^#^
**Positions of Dental Health Personnel (persons)**	Number (%)	Number (%)	1.000 ^##^
Dental Public Health Practitioner	2 (33.3%)	1 (16.7%)	
Senior Dental Public Health Practitioner	4 (66.7%)	4 (66.6%)	
Public Health Academic Officer	0 (0.0%)	1 (16.7%)	

Annotations: ^#^, analyzed using unpaired *t*-test statistics; ^##^, analyzed using exact test statistics. Abbreviations: SHPH, sub-district health promotion hospital; SD, standard deviation.

**Table 3 healthcare-13-00874-t003:** Comparison of the overall dental services for children aged 0–12 years between the groups of SHPH that were transferred and those that were not transferred during 2017–2021.

Dental Services	SHPH Status	Number of People Receiving Dental Services by Year (Persons)					
		2017	2018	2019	2020	2021	Mean
Oral health examinations for children aged 0–12 years	Transferred	1929	3061	3765	2881	2306	2788.4
	Non-transferred	2722	3284	4711	3197	1341	3051.0
Provision of topical fluoride application services for children aged 0-5 years	Transferred	601	1088	1014	1015	446	832.8
	Non-transferred	1543	1405	1346	1201	720	1243.0
Sealant for permanent teeth for children aged 6 years	Transferred	46	181	186	252	28	138.6
	Non-transferred	236	235	208	336	199	242.8

Abbreviations: SHPH, sub-district health promotion hospital.

**Table 4 healthcare-13-00874-t004:** Comparison of oral health conditions among children aged 0–12 years between transferred and non-transferred SHPHs during 2017–2021.

Oral Health Conditions by Children Age	SHPH Status	Year					
		2017	2018	2019	2020	2021	Mean
**Prevalence of dental caries (%)**							
3 years ^#^	Transferred	38.2	39.4	27.1	28.4	27.0	31.3
	Non-transferred	26.9	23.0	26.2	8.1	17.0	22.2
5 years ^#^	Transferred	28.8	47.4	56.7	62.4	70.1	51.2
	Non-transferred	25.1	39.1	37.1	25.6	13.5	30.7
12 years ^##^	Transferred	28.9	29.4	31.7	57.8	44.6	37.1
	Non-transferred	37.2	27.3	21.1	34.5	29.6	30.0
**The average number of decayed, filled, or extracted teeth (teeth per person)**							
3 years ^#^	Transferred	2.1	1.5	1.0	1.1	1.5	1.4
	Non-transferred	0.7	0.8	1.3	0.7	0.7	0.9
5 years ^#^	Transferred	1.4	2.5	2.4	3.6	3.9	2.6
	Non-transferred	1.9	3.2	2.3	1.9	1.3	2.3
12 years ^##^	Transferred	0.9	0.7	0.7	1.6	1.1	1.0
	Non-transferred	0.8	0.9	0.4	1.1	0.4	0.7

Annotations: ^#^, primary teeth or baby teeth; ^##^, permanent teeth or adult teeth. Abbreviations: SHPH, Sub-district health promotion hospital.

**Table 5 healthcare-13-00874-t005:** Comparison of oral health conditions for children aged 0–12 years between the groups of SHPH that were transferred and those that were not transferred during 2017–2021, using Chi-square test and unpaired *t*-test statistics.

Oral Health Conditions	Status		
	Transferred Group	Non-Transferred Group	
**Prevalence of dental caries**	**Number (%)**	**Number (%)**	***p*-Value**
**Children aged 3 years**			<0.001 ^#^
Number of people with detected dental caries	188 (31.3%)	187 (22.2%)	
Number of people without dental caries	412 (68.7%)	655 (77.8%)	
**Children aged 5 years**			<0.001 ^#^
Number of children with detected dental caries	283 (51.2%)	219 (30.7%)	
Number of children without dental caries	270 (48.8%)	495 (69.3%)	
**Children aged 12 years**			0.009 ^#^
Number of children with detected dental caries	208 (37.1%)	204 (30.0%)	
Number of children without dental caries	353 (62.9%)	475 (70.0%)	
**The average number of decayed, filled, or extracted teeth (teeth per person)**	**Mean (SD)**	**Mean (SD)**	**diff_mean_ ^##^ (95% CI)**
**Children aged 3 years** (primary teeth/baby teeth)	1.4 (2.3)	0.9 (1.9)	0.5 (0.3 to 0.7)
**Children aged 5 years** (primary teeth/baby teeth)	2.6 (2.8)	2.3 (4.0)	0.3 (−0.1 to 0.7)
**Children aged 12 years** (permanent teeth/adult teeth)	1.0 (1.4)	0.7 (1.5)	0.3 (0.1 to 0.5)

Annotations: ^#^, analyzed using Chi-square test statistics; ^##^, analyzed using unpaired *t*-test statistics. Abbreviations: SD, standard deviation; diff_mean_, mean difference.

## Data Availability

The data that support the findings of this study are available from the corresponding author, upon reasonable request.

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
