# Peer review of "Dental Service Provision and Oral Health Conditions of Children Aged 0–12 Years, Northern Thailand: Transferring of Sub-District Health Promotion Hospitals Policy Era"

_healthcare, 2025, doi:10.3390/healthcare13080874_

Round 1
Reviewer 1 Report
Comments and Suggestions for Authors
The abstract needs reformulation. Please reduce it and start with a background before the objective of the study.
Please change the keywords according to mesh terms and alphabetical order.
when you want to put more than 2 references it should be for example [6-9]. Please correct.
Please make the objective more clear and make it in the introduction part.
Figure 1. onceptual framework. please correct.
The key informants in this study were dental service providers stationed at Subdis- trict Health Promotion Hospitals (SHPHs) who meet the following criteria: Please put a reference for this criteria.
Results are clear.
for the discussion please state of your objective is accepted or rejected make a discussion according to previous articles.
Future directions should be implemented after the limitations.
For the conclusion. Make it in one paragraph.
Comments on the Quality of English Language
Moderate english
Author Response
Response to reviewer
Thank you very much for your fruitful comments. We revised as much as possible. We hope that it will satisfy the reviewer. However, please let us know if any need to be edited.
Reviewer 1
A: We first would like to declare that we decided to remove the qualitative part in this study because it was partially in the previous work (published in Thai); we do not want to let it be as a translated version. We do apologize for the reviewer. However, we confirm that the quantitative part is totally different in terms of data sources and analysis. Previously published article using data from health data center, but the current study uses the data form the sub-district health promotion hospitals. We also mentioned the previous work in the Methods.
Q1 The abstract needs reformulation. Please reduce it and start with a background before the objective of the study.
A: Agreed. We have reformulated and revised the abstract, especially we put the background before the objective. (Line 11: 15)
Q2 Please change the keywords according to mesh terms and alphabetical order.
A: Agreed. We changed the keywords to “Dental Health Services, Oral Health, Child, Local Government, Thailand” which match to the mesh terms.
Q3 when you want to put more than 2 references it should be for example [6-9]. Please correct.
A: Agreed. We have revised the reference throughout the manuscript.
Q4 Please make the objective clearer and make it in the introduction part.
A: Agreed. We revised the objective and also put it in the introduction part. The revised objective is “This study aimed to investigate and compare the dental services and oral health conditions of children aged 0-12 years between transferred and non-transferred SHPHs in northern Thailand”. (Line 69 to 71)
Q5 Figure 1. conceptual framework. please correct.
A: We decide to remove the conceptual framework, for we decide to keep just only quantitative part as mentioned earlier.
Q6 The key informants in this study were dental service providers stationed at Subdis- trict Health Promotion Hospitals (SHPHs) who meet the following criteria: Please put a reference for this criterion.
A: We decided to remove qualitative part as mentioned earlier.
Q7 Results are clear.
A: Thank you for your consideration.
Q8 for the discussion please state of your objective is accepted or rejected make a discussion according to previous articles.
A: Agreed. We state that our result follows the alternative hypothesis. We did try to search and compare our results with other related works, including the dental service provision (Line: 262 to 271). We alsocompare and discuss oral health conditions with other works. (Line 272 :296)
PS: There are quite limit to this field because the situation and the policy is unique in Thailand.
Q9 Future directions should be implemented after the limitations.
A: We have combined the limitation and future recommendation (Line: 297 to 307)
Q10 For the conclusion. Make it in one paragraph.
A: Agreed. We have revised it as “Transferred SHPHs have lower dental service provision but higher oral health problems than non-Transferred SHPHs. Therefore, the Ministry of Public Health (MOPH), at the CUP level, should revise resource allocation, including personnel, equipment, and supplies, to ensure fairness and consistency in service delivery for transferred SHPHs. Close monitoring for the dental service provision and oral health among children aged 0-12 years is needed, especially in area of responsibility of the transferred SHPHs”
Reviewer 2 Report
Comments and Suggestions for Authors
Dear respected authors: I have read your paper with interest and note this is an exciting field in Dental Public health. Overall, this is a well-written paper, and the results are significant for health care professionals. However, numerous studies have already been conducted in this area. The choice of this subject needs adequate study background in your community.
The title to be self-informative; the type of the study better to be determined;
Deficiencies exist in the statistical test results by reproducibility and reliability.
The data collection (Data sources) lacks adequate transparency;
- Reliable historical records such as medical charts, health databases, insurance records, etc.
- Quality of the data: comprehensive or missing (incomplete) information can lead to bias.
- Inclusion and exclusion criteria.
- Bias Consideration.
Deficiency exists in analysis of outcomes; it is important to calculate and interpret confidence intervals (CI) to understand the precision of the estimates.
Limitation of the applied methods (phase 2 and 1) not described adequately related to:
Data quality, Bias, Causality.
More detail in Data Collection Techniques is necessary related to phase 1.
Author Response
Response to reviewer 2
Thank you very much for your fruitful comments. We revised as much as possible. We hope that it will satisfy the reviewer. However, please let us know if any need to be edited.
A: We first would like to declare that we decided to remove the qualitative part in this study because it was partially in the previous work (published in Thai); we do not want to let it be as a translated version. We do apologize for the reviewer. However, we confirm that the quantitative part is totally different in terms of data sources and analysis. Previously published article using data from health data center, but the current study uses the data form the sub-district health promotion hospitals. We also mentioned the previous work in the Methods.
Q1 The title to be self-informative; the type of study better to be determined;
A: Agreed, we have revised the title as “Dental Service Provision and Oral Health Conditions of Children Aged 0-12 Years, Northern Thailand: Transferring of Sub-district Health Promotion Hospitals Policy Era”
Q2 Deficiencies exist in the statistical test results by reproducibility and reliability.
A: Agreed. We have added the 95%CI to the calculation and all number, mean and SD is sufficient to do inferential statistics.
Q3 The data collection (Data sources) lacks adequate transparency;
- Reliable historical records such as medical charts, health databases, insurance records, etc.
- Quality of the data: comprehensive or missing (incomplete) information can lead to bias.
- Inclusion and exclusion criteria.
- Bias Consideration.
A: Agreed. We have revised accordingly, as indicate in the 2.4. Data source and Data Collection Process section (Line 115 to 134).
Q4 Deficiency exists in analysis of outcomes; it is important to calculate and interpret confidence intervals (CI) to understand the precision of the estimates.
A: Agreed. We have revised the result and report with the CI in the estimate mean difference in Table 5.
Q5 Limitation of the applied methods (phase 2 and 1) not described adequately related to:
Data quality, Bias, Causality.
A: We have revised our limitations, including the limitations of this data source.
Q6 More detail in Data Collection Techniques is necessary related to phase 1.
A: We decide to remove the qualitative.
Reviewer 3 Report
Comments and Suggestions for Authors
Comparison of Dental Service Provision and Oral Health Conditions of Children Aged 0-12 Years between Transferred and Non-Transferred Sub-district Health Promotion Hospitals to Local Administrative Organizations in Northern Thailand
Reviewer Report
Thanks to authors for their study. Authors compared the dental services provided to children aged 0-12 in northern Thailand between transferred and non-transferred Subdistrict Health Promotion Hospitals (SHPHs). The study was conducted in two phases: the first phase examined the state of dental service provision, while the second phase analyzed data from 2017 to 2021. Overall, while dental service delivery inside and outside SHPHs was similar, minor differences were identified in the accreditation of dental hygienists, support for dental equipment, and assistance from dentists at affiliated hospitals. Transferred SHPHs provided fewer dental services compared to non-transferred SHPHs, and children in these areas exhibited higher rates of dental caries. The study revealed that transferred SHPHs had higher average scores for decayed, missing, and filled teeth. While the authors provide significant findings, there are aspects of the study that need further improvement:
- Four-line study title is too long; please shorten it.
- Present the abstract under the subheadings: Aim, Methods, Findings, and Conclusion. In addition, provide more specific and stronger recommendations in the conclusion section.
- Introduction is sufficient. It clearly presents the purpose, methodology, and conceptual framework of the study, effectively laying the foundation for the study.
- Materials and methods section is generally well-structured and carefully designed. It combines both qualitative and quantitative methods to provide in-depth and comprehensive analysis. However, more detailed information should be provided regarding the inclusion/exclusion criteria and sample size.
- Abbreviations used in the tables should be stated below the tables.
- Discussion section is brief and should be expanded and written in more detail. More literature should be searched to explain the factors causing differences in dental health services. Stronger recommendations or strategies should be added to address these issues. Based on the results, more concrete comments and suggestions should be provided.
- Combine the limitations with the Discussion and also highlight the strengths of the study.
- Include ‘7. Recommendations for Future Research’ section within the Discussion. There is no need for too many headings.
- Revise the references according to the journal’s guidelines.
Author Response
Response to reviewer 3
Thank you very much for your fruitful comments. We revised as much as possible. We hope that it will satisfy the reviewer. However, please let us know if any need to be edited.
A: We first would like to declare that we decided to remove the qualitative part in this study because it was partially in the previous work (published in Thai); we do not want to let it be as a translated version. We do apologize for the reviewer. However, we confirm that the quantitative part is totally different in terms of data sources and analysis. Previously published article using data from health data center, but the current study uses the data form the sub-district health promotion hospitals. We also mentioned the previous work in the Methods.
Reviewer Report
Thanks to authors for their study. Authors compared the dental services provided to children aged 0-12 in northern Thailand between transferred and non-transferred Subdistrict Health Promotion Hospitals (SHPHs). The study was conducted in two phases: the first phase examined the state of dental service provision, while the second phase analyzed data from 2017 to 2021. Overall, while dental service delivery inside and outside SHPHs was similar, minor differences were identified in the accreditation of dental hygienists, support for dental equipment, and assistance from dentists at affiliated hospitals. Transferred SHPHs provided fewer dental services compared to non-transferred SHPHs, and children in these areas exhibited higher rates of dental caries. The study revealed that transferred SHPHs had higher average scores for decayed, missing, and filled teeth. While the authors provide significant findings, there are aspects of the study that need further improvement:
Q1 Four-line study title is too long; please shorten it.
A: Agreed. We have revised accordingly. The revised title is “Dental Service Provision and Oral Health Conditions of Children Aged 0-12 Years, Northern Thailand: Transferring of Sub-district Health Promotion Hospitals Policy Era”
Q2 Present the abstract under the subheadings: Aim, Methods, Findings, and Conclusion. In addition, provide more specific and stronger recommendations in the conclusion section.
A: Agreed. We have revised the abstract subheadings and made a specific recommendation and conclusion that related to our findings.
Q3 Introduction is sufficient. It clearly presents the purpose, methodology, and conceptual framework of the study, effectively laying the foundation for the study.
A: Thank you very much for your complement.
Q4 Materials and methods section is generally well-structured and carefully designed. It combines both qualitative and quantitative methods to provide in-depth and comprehensive analysis. However, more detailed information should be provided regarding the inclusion/exclusion criteria and sample size.
A: Agreed. We added the information as suggested. However, we use the whole data set from the given period; therefore the 95%CI would reflect our precise of the estimation.
PS if we consider using the sample size calculation based on the two-independent T-test. It showed 105 sample needed.
Q5 Abbreviations used in the tables should be stated below the tables.
A: Agreed. We have put the footnote for abbreviations
Q6 Discussion section is brief and should be expanded and written in more detail. More literature should be searched to explain the factors causing differences in dental health services. Stronger recommendations or strategies should be added to address these issues. Based on the results, more concrete comments and suggestions should be provided.
A: We state that our result follows the alternative hypothesis. We did try to search and compare our results with other related works, including the dental service provision (Line: 262 to 271). We alsocompare and discuss oral health conditions with other works. (Line 272 :296)
PS: There are quite limit to this field because the situation and the policy is unique in Thailand.
Q7 Combine the limitations with the Discussion and highlight the strengths of the study. Include ‘7. Recommendations for Future Research’ section within the Discussion. There is no need for too many headings
A: Agreed. We combined the limitations and recommendations with the discussion and put strengths of the study (Line 297-307).
Q9 Revise the references according to the journal’s guidelines.
A: We have revised the reference to follow the journal's guidelines.
Reviewer 4 Report
Comments and Suggestions for Authors
Dear Authors,
Thank you for the opportunity to review your manuscript entitled "Comparison of Dental Service Provision and Oral Health Conditions of Children Aged 0-12 Years between Transferred and Non-Transferred Sub-district Health Promotion Hospitals to Local Administrative Organizations in Northern Thailand". It is interesting and can contribute to the scientific literature. My suggestions for improvement are listed below:
- The title seems too long. It should be direct and concise, focusing on the most important aspect of your manuscript.
- I suggest the authors introduce and explain the terms "sub-district health promotion hospitals" and "local administrative organizations" at the beginning of the Introduction sections to make the manuscript comprehensible for readers unfamiliar with the topic.
- I suggest that lines 91-102 be included in the Materials and Methods section.
- Please reorganize the sub-sections of the Materials and Methods more concisely, e.g., report both phases in the "Research Methodology" (2.2 "Population and Sample", 2.2.1. Phase 1; 2.2.2 Phase 2, and so on) OR 2.1. Phase 1 (2.1.1. Population and Sample; 2.1.2. Key Informants....) and 2.2. Phase 2 with the corresponding sub-sections.
- I suggest the authors emphasize the study's strengths at the end of the Discussion, report its limitations in the same section (could be as a sub-section), and give their recommendations for further research (again as a part of the Discussion).
- The Conclusions should summarize the key findings and recommendations. It can be extended.
- Please, strictly follow the Instructions for Authors and format the text and references in accordance with them.
Author Response
Response to reviewer 4
Thank you very much for your fruitful comments. We revised as much as possible. We hope that it will satisfy the reviewer. However, please let us know if any need to be edited.
A: We first would like to declare that we decided to remove the qualitative part in this study because it was partially in the previous work (published in Thai); we do not want to let it be as a translated version. We do apologize for the reviewer. However, we confirm that the quantitative part is totally different in terms of data sources and analysis. Previously published article using data from health data center, but the current study uses the data form the sub-district health promotion hospitals. We also mentioned the previous work in the Methods.
Thank you for the opportunity to review your manuscript entitled "Comparison of Dental Service Provision and Oral Health Conditions of Children Aged 0-12 Years between Transferred and Non-Transferred Sub-district Health Promotion Hospitals to Local Administrative Organizations in Northern Thailand". It is interesting and can contribute to the scientific literature. My suggestions for improvement are listed below:
Q1 The title seems too long. It should be direct and concise, focusing on the most important aspect of your manuscript.
A: Agreed, we have revised the title as “Dental Service Provision and Oral Health Conditions of Children Aged 0-12 Years, Northern Thailand: Transferring of Sub-district Health Promotion Hospitals Policy Era”
Q2 I suggest the authors introduce and explain the terms "sub-district health promotion hospitals" and "local administrative organizations" at the beginning of the Introduction sections to make the manuscript comprehensible for readers unfamiliar with the topic.
A: Agreed. We have brief explain about the sub-district health promotion hospitals" and "local administrative organizations" at the beginning of the Introduction. (Line 37:48)
Q3 I suggest that lines 91-102 be included in the Materials and Methods section.
A: Thank you very much. We have revised accordingly.
Q4 Please reorganize the sub-sections of the Materials and Methods more concisely, e.g., report both phases in the "Research Methodology" (2.2 "Population and Sample", 2.2.1. Phase 1; 2.2.2 Phase 2, and so on) OR 2.1. Phase 1 (2.1.1. Population and Sample; 2.1.2. Key Informants....) and 2.2. Phase 2 with the corresponding sub-sections.
A: Thank you very much. We have revised accordingly.
Q5 I suggest the authors emphasize the study's strengths at the end of the Discussion, report its limitations in the same section (could be as a sub-section), and give their recommendations for further research (again as a part of the Discussion).
A: Thank you very much. We totally agreed. We revised it as stated in Line (263 – 275)
Q6 The Conclusions should summarize the key findings and recommendations. It can be extended.
A: Agreed. We have revised the conclusion based on our findings and provide recommendation accordingly.
Q7 Please, strictly follow the Instructions for Authors and format the text and references in accordance with them.
A: we have check and revised accordingly.
Round 2
Reviewer 2 Report
Comments and Suggestions for Authors
Dear authors:You have adequately addressed the majority of the concerns raised during the initial review. Your revisions have significantly improved the manuscript, and the responses to the reviewers’ comments are thorough and well-considered. There is no need for extra round of review.
Author Response
Thank you very much for your positive feedback. I appreciate your time and consideration in reviewing my manuscript. I’m glad to hear that the revisions were satisfactory
Reviewer 3 Report
Comments and Suggestions for Authors
Thank the authors for their revisions. The study has improved to a publishable level. I have no additional comments.
Author Response

(The authors gave the same response as above.)

Reviewer 4 Report
Comments and Suggestions for Authors
The manuscript has been substantially improved. I suggest the authors replace "our research/study" with "this/the present research/study".
Author Response
The manuscript has been substantially improved. I suggest the authors replace "our research/study" with "this/the present research/study".
Response:
Thank you very much for your suggestions.
Agreed. Throughout the manuscript, we have replaced "our research/study" with "this/the present research/study."